# Chemotaxis to plant defense compounds in phytopathogens

Roberta Genova[1,2], Ashleigh Holmes[3], Mario Cano-Muñoz[1], Atsushi Ishihara[4], Naoki Ube[5], Taiji Nomura[5], Jose A. Gavira[6], Miguel A. Matilla[1]*, Tino Krell [1]*

1 Department of Biotechnology and Environmental Protection, Estación Experimental del Zaidín, CSIC, Granada, Spain, 2 PhD program in Fundamental and Systems Biology, Granada University, Granada, Spain, 3 Department of Cell and Molecular Science, James Hutton Institute, Dundee, Scotland, United Kingdom, 4 Faculty of Agriculture, Tottori University, Tottori, Japan, 5 Biotechnology Research Center and Department of Biotechnology, Toyama Prefectural University, Toyama, Japan, 6 Laboratory of Crystallographic Studies, Instituto Andaluz de Ciencias de la Tierra, CSIC, Armilla, Spain

* miguel.matilla@eez.csic.es (MAM); tino.krell@eez.csic.es (TK)

## Abstract

Plant pathogens possess about twice as many chemoreceptors as the bacterial average, suggesting broad chemotactic capacities. The signals recognized by most phytopathogen chemoreceptors are unknown, and the reasons for this elevated chemoreceptor number is unclear. We identified the signals recognized by three chemoreceptors, PacH, PacI and PacG, in the global phytopathogen *Pectobacterium atrosepticum*. The ligand-binding domains (LBDs) of these chemoreceptors share modest sequence similarity, but the signals they recognize are structurally similar, and their biosynthetic pathways are interwoven. Whereas PacH and PacI recognized benzoate derivatives, including salicylate, vanillin and *p*-hydroxybenzoate, PacG bound agmatine, feruloylagmatine and *p*-coumaroylagmatine. These compounds are known plant defense compounds, their production is induced by pathogen attack, and they typically accumulate at infection sites. All compounds, except agmatine, induced chemoattraction, which was abolished by mutations in the corresponding genes. Agmatine competed with feruloylagmatine and *p*-coumaroylagmatine for PacG-LBD binding *in vitro* and antagonized chemotaxis *in vivo*. A mutant in *pacG*, but not in other chemoreceptor genes, showed reduced virulence *in planta*. We report high-resolution structures of PacG-LBD that were used for ligand-docking experiments to identify its binding pocket. PacH, PacI and PacG homologs were identified in other important phytopathogens belonging to the *Burkholderia, Erwinia, Ralstonia, Pectobacterium* and *Dickeya* genera. This is the first report of chemotaxis to feruloy- lagmatine, *p*-coumaroylagmatine and *p*-methoxybenzoate, expanding the range of chemoeffectors. Bacteria thus exploit plant defense responses by moving to com- pounds that are secreted at infection sites in response to pathogen attack. Chemo- taxis to plant defense compounds may be a means to access infected plants and infection sites.

**Data availability statement:** Data are deposited at Genova, R., Matilla, M. A., & Krell, T. (2026). Chemotaxis to Plant Defense Compounds in Phytopathogens. Zenodo. https://doi.org/10.5281/zenodo.18660724 (DOI: 10.5281/zenodo.18660724).

**Funding:** This study was supported by grants from the Spanish Ministry for Science, Innovation and Universities/Agencia Estatal de Investigación 10.13039/501100011033 and FEDER, UE through grants PID2020-112612GB-I00 and PID2023-146216NB-I00 to TK, grants PID2023-146281NB-I00 and PID2019-103972GA-I00 to MAM, the Juan de la Cierva postdoctoral fellowship JDC2022-049681-I and the predoctoral grant PRE2021-096885 to MCM and RG, respectively. We acknowledge EMBO for the Scientific Exchange Grant 10969 and the Granada University for predoctoral training of RG. AH was supported by the Scottish Government Environment, Natural Resources and Agriculture Research Program (2022-2027) under project JHI-A1-1, Epidemiology of Key Pests and Diseases. NU was supported by the JSPS KAKENHI Grant 24K17838. The funders had no role in study design, data collection and analysis, decision to publish, or preparation of the manuscript.

**Competing interests:** The authors have declared that no competing interests exist.

## Author summary

Chemotaxis permits bacteria to move to sites that are favorable for survival and host colonization. A typical chemotactic response is initiated by the binding of chemoeffectors to chemoreceptors. Plant-associated bacteria possess about twice as many chemoreceptors as the average bacterium, suggesting that many of these receptors mediate chemotaxis to plant-derived compounds. However, proof of this notion is lacking since the signals recognized by the very large majority of chemoreceptors are unknown. We show here that three chemoreceptors (PacG, PacH and PacI) of the global phytopathogen *Pectobacterium atrosepticum* bind different plant defense compounds, mediating chemoattraction. These compounds are known to be secreted by plants following pathogen attack. Chemoreceptor PacG bound specifically feruloylagmatine and *p*-coumaroylagmatine, which are novel chemoeffectors. The deletion of the *pacG* gene, but not that of other so far characterized *P. atrosepticum* chemoreceptors, reduced virulence in potato tubers and impaired plant entry through wounds. We report the high-resolution structure of the PacG ligand binding domain that may serve to identify receptor homologs in other bacteria. Chemotaxis to plant defense compounds may be a means to access infected plants and infection sites, leading to a potentiation of bacterial infection.

## Introduction

Chemotaxis is the directed movement along chemical gradients that allows bacteria to reach sites that are favorable for survival and host colonization [1,2]. The primary benefit of chemotaxis is to access nutrients [1,3]. However, chemotaxis also occurs in response to specific environmental cues, often with no metabolic value, that provide bacteria with information about their environment or warn of environmental stresses [1,3]. In this respect, chemotaxis to such cues modulates the interaction among bacteria of the same [4,5] or different species [6,7]. Chemotaxis to specific environmental cues is also required for colonization of humans/animals [8,9], plants [10,11] or corals [12,13]. We are only beginning to explore the spectrum of environmental cues and the physiological role of the resulting chemotactic movement.

Bacteria that associate with hosts possess a higher number of chemoreceptors than the bacterial average [14,15], suggesting that many of these receptors might sense host compounds. Plant-pathogenic bacteria, for example, possess an average of 27 chemoreceptors, compared to the bacterial average of 12 [15]. The importance of chemotaxis in plant colonization is illustrated by the observation that the mutation of chemotaxis genes reduces plant colonization by different beneficial [16–18] and pathogenic bacteria [19–21]. However, our knowledge of the plant compounds that attract bacteria is poor: the signal recognized by most chemoreceptors is unknown [22].

A chemotactic response is typically initiated when a chemoeffector binds to the ligand-binding domain (LBD) of a chemoreceptor [23]. The ligand specificity of a

chemoreceptor is determined by its LBD. Sequence analyses of chemoreceptor genes also indicate that there are many receptor families that specifically recognize plant compounds [15]. Based on an alignment of all available chemoreceptor LBDs, they can be grouped into about 5,000 clusters [15]. For each cluster, the relative abundance of receptors originating from plant-associated bacteria, including phytopathogens, was assessed and expressed as the degree of plant specificity (DPS). Many clusters were predominantly composed of LBDs of receptors from plant-associated bacteria [15], suggesting that cluster members may respond to plant-derived compounds that are, in almost all cases, unknown.

To study chemoreceptor function, we used *Pectobacterium atrosepticum,* which is among the top ten most relevant bacterial plant pathogens [24]. The reference strain SCRI1043 has 36 chemoreceptors that interact with a single chemotaxis pathway [25]. To date, we have identified chemoreceptors that respond to formate [26], nitrate [27], quaternary amines [28], amino acids [29] and phosphorylated C3 compounds [30]. Chemoreceptor genes are typically scattered throughout the genome [31]. However, we were attracted by a cluster of four chemoreceptor genes. Here, we identify the signals recognized by three of these chemoreceptors and analyze their function. These receptors respond to different plant defense compounds. Using defined chemoreceptor mutants, we evaluated the impact of the chemotaxis response to these phytochemicals on bacterial plant pathogenicity. Our data indicate that bacteria exploit the defense mechanisms of plants to induce chemotaxis to infection sites.

## Results

### SCRI1043 encodes a cluster of four chemoreceptor genes that are not co-transcribed

Of the 36 chemoreceptors of *P. atrosepticum* SCRI1043, we noted that 4 of them, ECA_RS21440, ECA_RS211445, ECA_RS21450, and ECA_RS21455, were encoded consecutively at one genetic locus, which is uncommon for chemoreceptor genes (Fig 1A).

LBD sequence-cluster analysis shows that the LBDs of these receptors possess elevated DPS scores, indicating that they are primarily found in plant-associated bacteria (Fig 1A, S1 Table). When we inspected other members of these LBD clusters (S2 Table), we noted that many are from important plant pathogens that belong to the genera *Burkholderia, Erwinia, Ralstonia, Pectobacterium* and *Dickeya*, suggesting that they may share the same function. These chemoreceptors possess four-helix bundle type LBDs, but pairwise alignments show only modest amino acid sequence identities of 26–51% (S3 Table).

To elucidate whether these genes form an operon, we employed a reverse transcriptase PCR approach. We designed primer pairs that amplify across the untranslated regions (UTRs) between neighboring genes (S1 Fig). In initial experiments, we conducted PCR experiments with the genomic DNA and observed the three corresponding bands at the

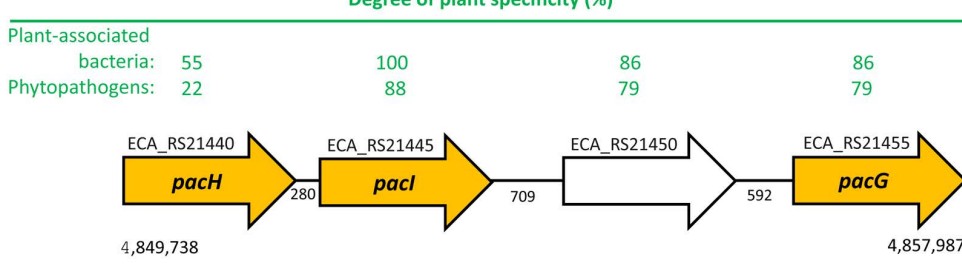

**Fig 1. The gene cluster encoding four chemoreceptors in *P. atrosepticum* SCRI1043.** Schematic representation of the *pacH-pacG* locus. The Degree of Plant Specificity (DPS) scores as reported by [15] are indicated. The DPS score corresponds to the abundance of chemoreceptors from plant-associated bacteria (PAB) and phytopathogens in each of the clusters, with a score of 100 indicating that all sequences of a given cluster are from PAB/phytopathogens and a score of 0 indicating that none are. The DPS scores of all *P. atrosepticum* SCRI1043 chemoreceptors are shown in S1 Table. The strains present in the clusters that contain PacH, PacI and PacG are provided in S2 Table. The numbers between the arrows represent the intergenic distances between adjacent genes.

expected sizes (S1 Fig). When these experiments were conducted with reverse-transcribed DNA (cDNA), a PCR band was not observed for any primer pair. When these experiments were repeated with primer pairs that amplify internal gene segments in cDNA, bands were observed in all cases (S2 Fig), indicating that these genes result in monocistronic transcripts.

## Chemoreceptor ECA_RS21440 (PacH) recognizes salicylate and vanillin

To identify ligands that activate these four chemoreceptors, their individual LBDs were overexpressed in *Escherichia coli* and purified by affinity chromatography. Three LBDs could be obtained as soluble proteins, but we were unable to identify a buffer system that guaranteed the stability of ECA_RS21450-LBD. The three soluble proteins were submitted to thermal-shift-based ligand-screening experiments using the PM1, PM2A, PM3B, PM4A and PM5 compound arrays comprising 475 compounds corresponding to different carbon, nitrogen, phosphorous, and sulphur sources and nutrients (S3 Fig). In these assays, protein-compound mixtures are exposed to a temperature gradient and protein unfolding is recorded. The interaction of a protein with a ligand typically delays thermal unfolding that can be quantified as an increase in the midpoint temperature of the unfolding transition (Tm) [32]. This screening with ECA_RS21440-LBD produced only a minor shift of 1.7 °C for salicylate (*o*-hydroxybenzoate). Isothermal titration calorimetry (ITC) experiments showed that salicylate bound to this LBD with negative cooperativity with a $K_{D1}$ value of 314 µM (Table 1, S4 Fig). Since the thermal shift assay can result in false negative results [33], we considered this compound as a lead and explored the binding of structurally related compounds using ITC. The protein was titrated with 24 compounds, most of which show structural similarity with salicylate (S4 Table). Of these, only vanillin showed binding ($K_{D1}$ = 66 µM, Table 1, S4 Fig). Salicylate is a major defense hormone in plants against a broad spectrum of pathogens, and its biosynthesis is induced by pathogen attack [34,35]. Vanillin has antibacterial [36,37] and antifungal [38–40] activities. Vanillin biosynthesis is induced in plant cells by salicylate [41]. This chemoreceptor was named PacH (*Pectobacterium atrosepticum chemoreceptor H*).

## Chemoreceptor ECA_RS21445 (PacI) recognizes salicylate and other benzoate derivatives

ECA_RS21445-LBD shares only 26% amino acid sequence identity with PacH-LBD (S3 Table). Ligand screening with the same compound arrays revealed significant increases in Tm for capric and salicylic acids (Fig 2A).

**Table 1. Dissociation constants derived from microcalorimetric binding studies to the ligand binding domains of chemoreceptors PacH, PacI and PacG and minimal inhibitory concentrations (MIC).**

| Protein | Ligand | Dissociation constant (µM)[a] | MIC value (mM) |
|---|---|---|---|
| PacH-LBD (ECA_RS21440-LBD) | Salicylate (*o*-hydroxybenzoate) | 314/1400 | 5 |
| | Vanillin | 66/3100 | 6 |
| PacI-LBD (ECA_RS21445-LBD) | Salicylate | 352/2700 | 5 |
| | Benzoate | 1300/23000 | >50 |
| | *p*-Chlorobenzoate | 19/85 | 2 |
| | *p*-Methoxybenzoate | 40/7300 | 20 |
| | *p*-Hydroxybenzoate | 42/172 | >50 |
| PacG-LBD (ECA_RS21455-LBD) | Agmatine | 37/1000 | >50 |
| | Feruloylagmatine | 28/1200 | >5[b] |
| | *p*-Coumaroylagmatine | 39/600 | 5 |

[a] Binding occurs with negative cooperativity. Provided are both constants ($K_{D1}/K_{D2}$)

[b] Maximal concentration

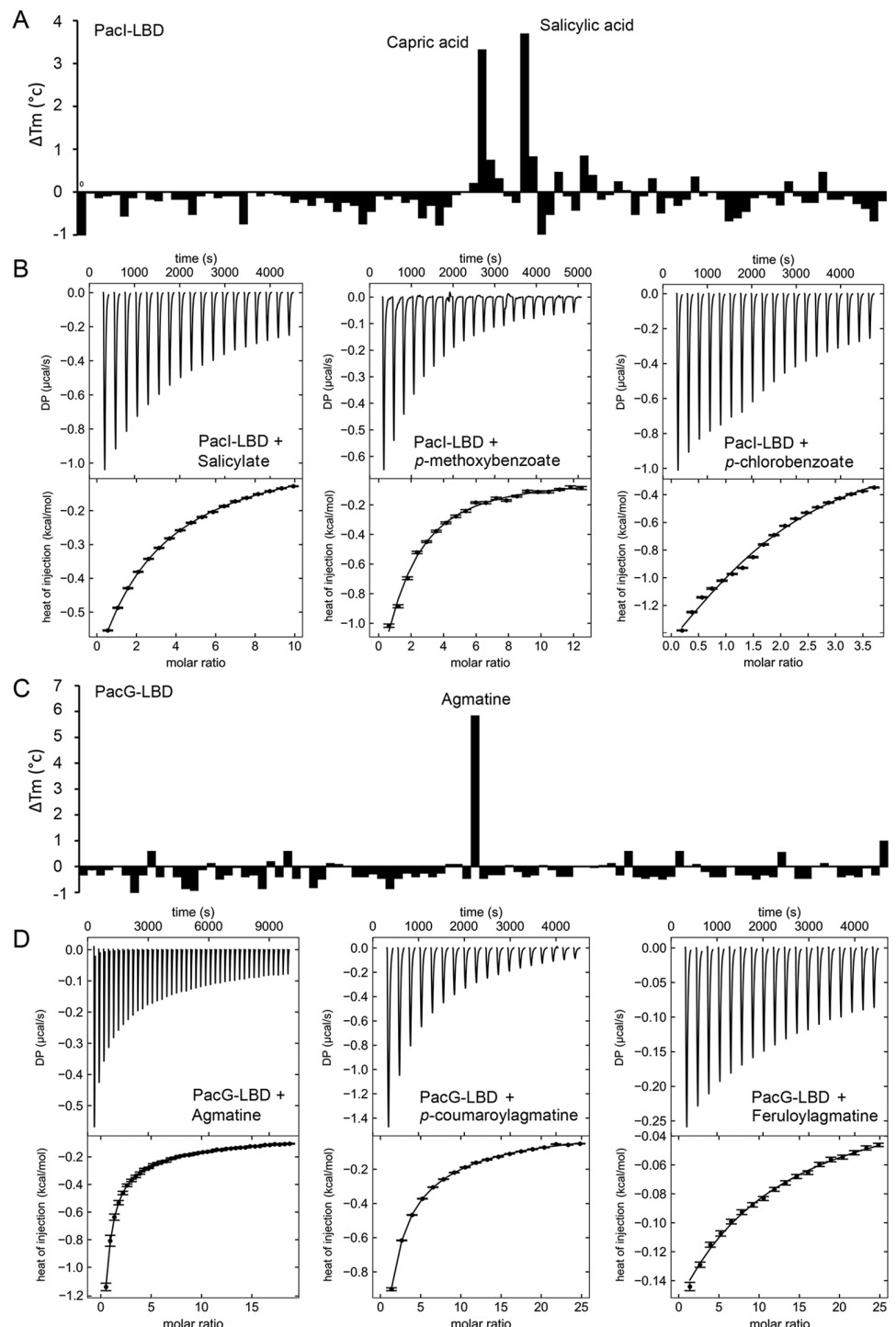

**Fig 2. Ligand binding to ECA_RS21445-LBD (PacI) and ECA_RS21455-LBD (PacG). A)** Thermal-shift assays with PacI-LBD; changes in Tm by the compounds of the Biolog compound array PM2A are shown. **B)** Microcalorimetric titration of 100 μM PacI-LBD with 12.8 μL aliquots of 5 mM salicylate**,** 2 mM *p*-methoxybenzoate, and 2 mM *p*-chlorobenzoate. **C)** Thermal-shift assays with PacG-LBD; changes in Tm by the compounds of the Biolog compound array PM3B are shown. **D)** Microcalorimetric titration of 32 to 100 μM PacG-LBD with 6.4 μL aliquots of 3 mM agmatine, and 14.4 μL aliquots of 5 mM *p*-coumaroylagmatine and feruloylagmatine. **B,D)** Upper panels: Raw titration data. Lower panels: Concentration-normalized and dilution heat-corrected integrated raw data. The derived dissociation constants are given in Table 1.

Subsequent microcalorimetric titrations showed binding of salicylate (Fig 2B, Table 1), but not capric acid (S5 Fig), indicating that the latter compound binds with very low affinity. Analogous to PacH-LBD, we used ITC to explore the binding of 25 structurally related compounds (S4 Table). ECA_RS21445-LBD failed to bind vanillin. A weak interaction was observed with benzoate ($K_{D1}$ = 1300 μM), but interactions with significantly higher affinities ($K_{D1}$ of 19–42 μM) were obtained for three para-substituted benzoate derivatives: p-chlorobenzoate, p-methoxybenzoate and p-hydroxybenzoate (Fig 2B, Table 1). p-hydroxybenzoate is among the most important plant defense molecules against bacterial infection. It has been detected in many plants [42]; it is present in important quantities in root exudates [43,44]; it is frequently toxic to pathogens [45]; and it is synthesized in response to plant infection [46]. Activity in plant defense has also been observed for benzoate and p-methoxybenzoate [45,47]. This chemoreceptor was named PacI (Pectobacterium atrosepticum chemoreceptor I).

### The plant defense phenolics feruloylagmatine and p-coumaroylagmatine are ligands of ECA_RS21455 (PacG)

ECA_RS21455-LBD shares 34 and 26% sequence identity with PacH-LBD and PacI-LBD (S3 Table), respectively. Ligand screening of ECA_RS21455-LBD with the thermal-shift assay using the same five compound arrays showed that only a single compound, agmatine, caused a significant increase in Tm (Fig 2C). Microcalorimetric titrations confirmed binding with a $K_{D1}$ of 37 μM (Table 1, Fig 2D). Agmatine is the decarboxylation product of arginine and is, in contrast to the PacH-LBD and PacI-LBD ligands, part of general metabolism and not specifically a defense compound. This finding, however, contrasts with the elevated DPS score of the receptor (Fig 1A), indicating that this receptor family is almost exclusively found in plant-associated bacteria. Published literature revealed that, in plants, agmatine is conjugated to ferulic and p-coumaric acid, giving rise to feruloylagmatine and p-coumaroylagmatine [48], two hydroxy-cinnamic acid amides (HCAAs). In contrast to agmatine, both compounds are known plant defense compounds [48,49]. To assess whether this chemoreceptor recognizes feruloylagmatine and p-coumaroylagmatine, we synthesized both compounds and verified their identity by NMR spectroscopy (S6 Fig). Microcalorimetric titration showed that both compounds bind to ECA_RS21455-LBD (Fig 2D) with affinities comparable to those of agmatine (Table 1). Titrations with ferulic and p-coumaric acid and other related compounds (S4 Table) did not show binding, indicating that the agmatine moiety is crucial for binding. The infection of plants with fungi [50,51] and bacteria [52] was shown to increase the local accumulation of feruloylagmatine and p-coumaroylagmatine. This chemoreceptor was named PacG (Pectobacterium atrosepticum chemoreceptor G).

### PacH and PacI ligands are of no apparent metabolic value

Although the LBDs of PacH, PacI and PacG share only a modest degree of sequence identity (S3 Table), their ligands share structural similarities, and their biosynthesis is interconnected (Fig 3A).

Importantly, all ligands of the three chemoreceptors are plant defense compounds, many of which are toxic to bacteria. There is extensive evidence that the PacH, PacI and PacG ligands possess antimicrobial activities against different bacteria [37,49,56]. To assess their antibacterial activity towards P. atrosepticum, we determined the minimal inhibitory concentrations (MIC) of all receptor ligands. No toxic effects at concentrations of up to 50 mM were detected for benzoate, p-hydroxybenzoate and agmatine, whereas the remaining compounds showed MIC values between 2–20 mM, indicative of moderate toxicity (Table 1). To assess the metabolic value of the PacH, PacI and PacG ligands, we conducted growth experiments in minimal medium supplied with the ligands as sole carbon or nitrogen source. Ligand concentrations were chosen to be well below the corresponding MIC values. Growth in the presence of glucose and ammonium sulfate served as a control (Fig 3B, 3C). We observed that the three PacG ligands, agmatine, p-coumaroylagmatine and feruloylagmatine, could serve as the sole N-source (Fig 3B), whereas only feruloylagmatine served as C-source (Fig 3C). Importantly, none of the PacH and PacI ligands were of metabolic value, suggesting that the functional relevance of both receptors is for detecting specific environmental cues rather than for gaining access to nutrients.

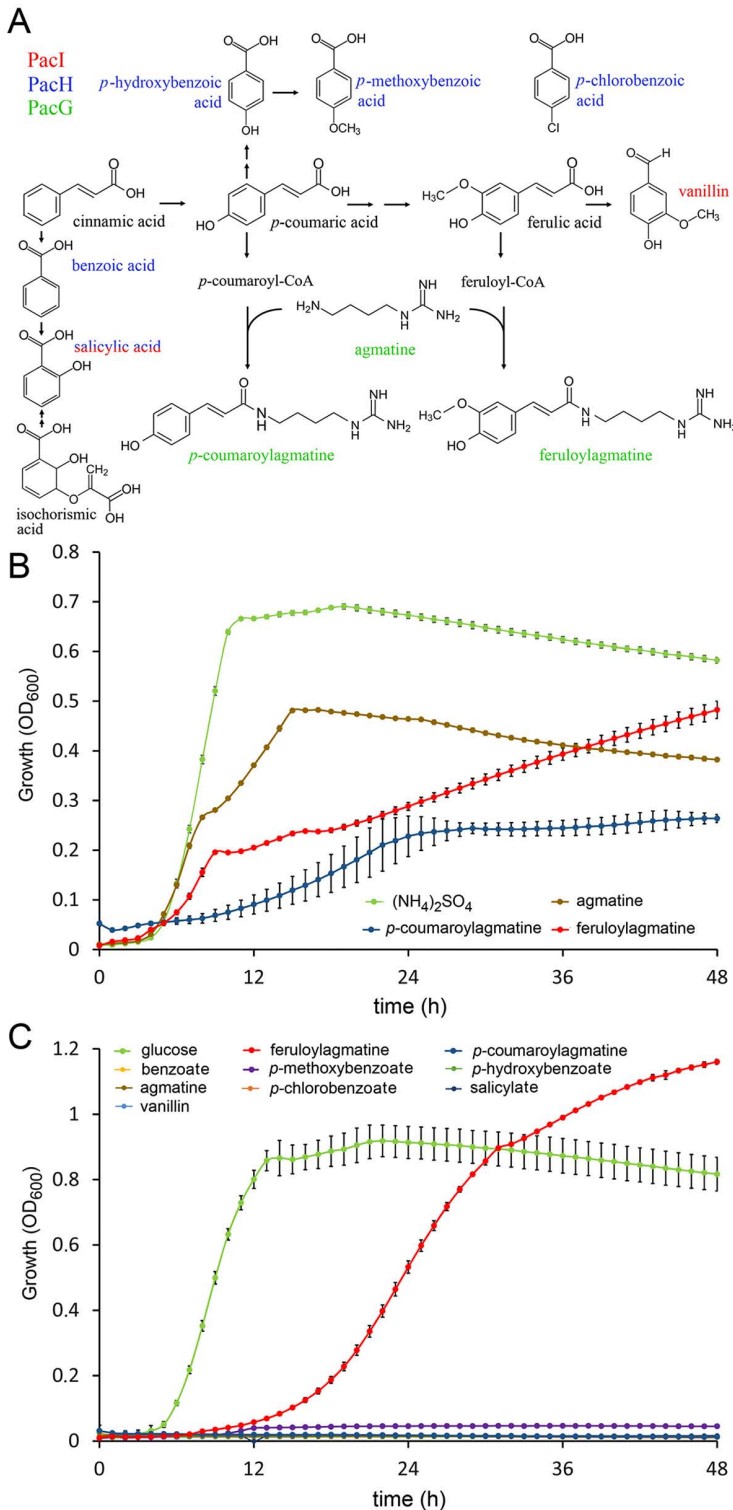

**Fig 3. The ability of ligands of chemoreceptors PacH, PacI and PacG to support growth of *P. atrosepticum* SCRI1043. A)** Structures and biosynthetic routes, adapted from [46,53–55]. **B, C)** Growth of *P. atrosepticum* SCRI1043 in minimal medium supplemented with the receptor ligands as sole nitrogen (**B**) or carbon source **(C)**. Growth in the presence of 0.1% (w/v) $(NH_4)_2SO_4$ and 0.2% (w/v) glucose as sole N- and C-source served as controls. Ligands were used at the following concentrations: 5 mM of *p*-methoxybenzoate, benzoate, *p*-hydroxybenzoate, agmatine, feruloylagmatine; 1 mM for the remaining ligands. Data are the means and standard deviations from three biological replicates.

## PacH and PacI ligands induce chemoattraction

To monitor chemotaxis of SCRI1043 to the ligands identified above, we conducted quantitative capillary chemotaxis assays. Initial experiments conducted in standard conditions, involving bacterial cell culture in minimal medium supplemented with glucose as sole carbon source, did not show any taxis to vanillin, salicylate or benzoate (S7 Fig). Chemotaxis can sometimes be induced by the presence of the cognate ligands in the culture medium [57], so we next monitored chemotaxis to vanillin, salicylate and benzoate in cultures grown in minimal medium supplemented with vanillin or benzoate. Whereas no significant chemotaxis was observed for cells grown in the presence of vanillin, growth with benzoate caused large increases in the chemotaxis to all three chemoeffectors (S7 Fig). In a control experiment, we show that benzoate does not alter chemotaxis to L-Asp that is mediated by the PacC chemoreceptor (S8 Fig) [29], indicative of a specific induction.

Using cultures grown in the presence of 0.5 mM benzoate, we conducted dose-response chemotaxis experiments using chemoeffectors at concentrations between 0.5 µM to 50 mM. We observed strong chemotaxis towards vanillin between 5–50 mM, whereas chemotaxis to salicylate led to 5–7 times lower accumulation that was most pronounced at lower concentrations of 0.5 to 1 µM (Fig 4A). SCRI1043 chemotaxis to the PacI ligands *p*-methoxybenzoate and *p*-hydroxybenzoate was assessed, with strong responses seen at 1 µM and 100 µM, respectively (Fig 4B). Chemotaxis to 50 mM vanillin has been observed (Fig 4), which corresponds to a concentration that is toxic to the cell (Table 1). This suggests that chemotaxis to plant defense compounds is a trade-off between the benefits arising from the attraction to these compounds and their toxic effects.

To assess the contribution of both chemoreceptors to these responses, we constructed defined *pacH* and *pacI* deletion mutants and measured the respective chemoattractant dose response. As shown in Fig 4A and 4B, the chemotaxis of both mutants to the benzoate derivatives was either abolished or largely reduced. Control experiments showed that responses of both mutants to L-Asp are comparable to that of the wild-type (WT) strain (S9A Fig).

## Agmatine antagonizes chemoattraction to *p*-coumaroylagmatine and feruloylagmatine

In contrast to the PacH and PacI ligands, chemotaxis to *p*-coumaroylagmatine and feruloylagmatine was observed in the absence of an inducer (Fig 4C). This is the first report of bacterial chemotaxis to both compounds [58]. Responses were strongest at concentrations between 50–500 µM. Importantly, no significant chemotaxis to agmatine was observed over the entire concentration range tested (10 µM to 10 mM), either in the absence or presence of agmatine in the culture medium (S10 Fig). A *pacG* deletion mutant was constructed, and control experiments showed that its responses to casamino acids were comparable to those of the WT strain (S9B Fig). Responses of this mutant to *p*-coumaroylagmatine and feruloylagmatine were either abolished or largely reduced (Fig 4C), indicating that PacG is the primary chemoreceptor for both compounds.

Ligand binding to receptor LBDs typically triggers downstream signaling. However, there are reports of ligands, termed antagonists, that bind to sensor kinases [59,60], transcriptional regulators [61,62] or chemoreceptors [63,64] but do not trigger a response. These compounds compete with agonistic ligands for binding at the receptor, reducing the magnitude of response [63,64]. To determine whether agmatine is an antagonist, we conducted *in vitro* microcalorimetric competition assays (Fig 5A).

We titrated PacG-LBD with *p*-coumaroylagmatine in the absence and presence of two different agmatine concentrations. In the presence of 200 µM agmatine, binding enthalpies were greatly reduced compared to titration in the absence of agmatine. Furthermore, in the presence of 2 mM agmatine, heat changes were indistinguishable from ligand dilution heats, indicating that agmatine and *p*-coumaroylagmatine compete for binding at PacG-LBD (Fig 5A).

We subsequently conducted *in vivo* chemotaxis competition assays and studied responses to *p*-coumaroylagmatine and feruloylagmatine in the absence and presence of two different agmatine concentrations (Fig 5B). In the presence of

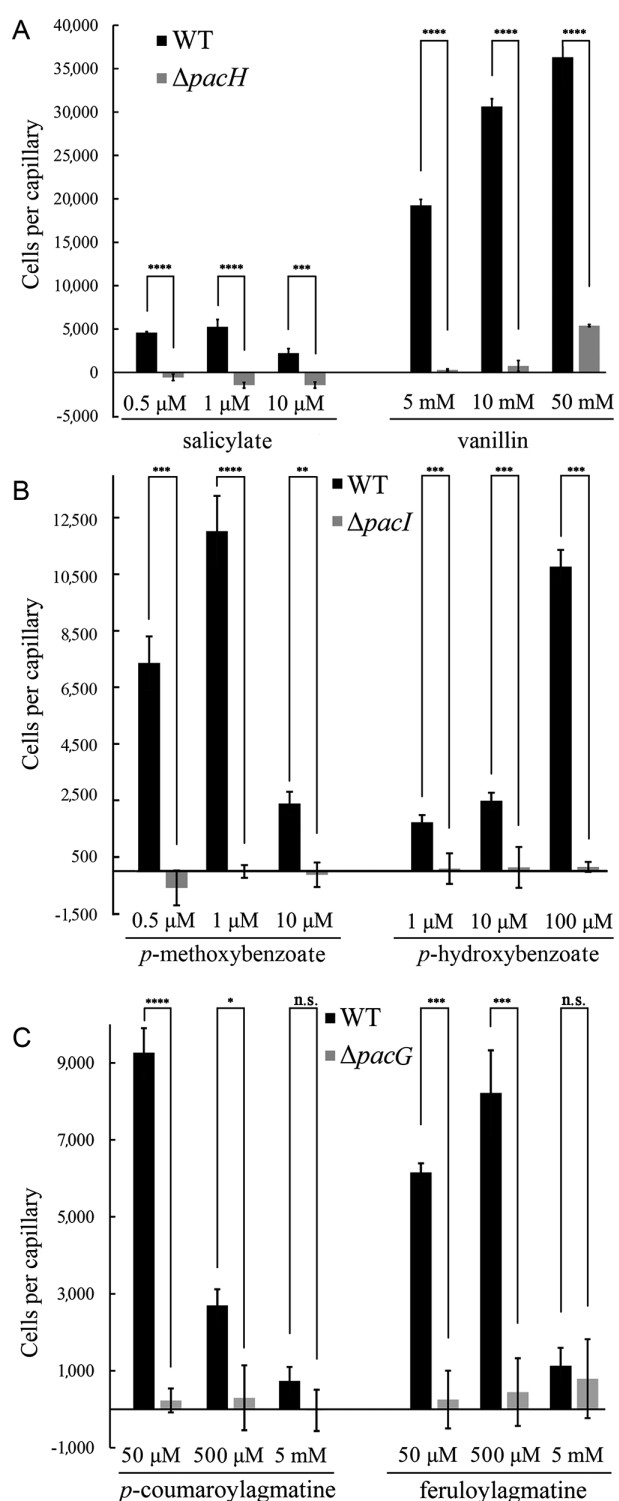

**Fig 4. Quantitative capillary assays of *P. atrosepticum* SCRI1043 and mutants deficient in the *pacH* (A), *pacI* (B) and *pacG* (C) genes towards different ligands.** For experiments shown in panels A and B, cells were grown in minimal medium supplemented with 20 mM glucose and 0.5 mM benzoate. For experiments of panel C, cells were grown in minimal medium containing 20 mM glucose. Data have been corrected for the number of cells that swam into buffer-only capillaries: 2,324 (WT salicylate), 2,771 (WT vanillin), 6,708 (ΔpacH salicylate), 5,375 (ΔpacH vanillin), 2,077 (WT *p*-methoxybenzoate), 2,372 (WT *p*-hydroxybenzoate), 1,567 (ΔpacI *p*-methoxybenzoate), 2,392 (ΔpacI *p*-hydroxybenzoate), 665 (WT *p*-coumaroylagmatine), 790 (WT feruloylagmatine), 1,489

(Δ*pacG* p-coumaroylagmatine), and 919 (Δ*pacG* feruloylagmatine). The means and standard deviations from three biological replicates conducted in triplicate are shown. Unpaired *t*-test: * p<0.05, ** p<0.005, *** p<0.0005, **** p<0.0001, n.s. not significant.

20 µM agmatine, responses were reduced by about 70%, and this response was further reduced in the presence of 200 µM agmatine (Fig 5B). These findings show that agmatine competes with *p*-coumaroylagmatine and feruloylagmatine for binding at PacG-LBD and antagonizes chemotaxis to both chemoeffectors.

## The PacG chemoreceptor is required for efficient virulence

To assess the role of PacH, PacI, and PacG chemoreceptors in plant virulence, several assays were conducted. First, we evaluated the capacity of different strains to cause soft rot in the potato tuber. To evaluate the role of chemotaxis in *P. atrosepticum* virulence, pathogenicity experiments were conducted with a non-chemotactic strain, Δ*cheA,* which lacks the central histidine kinase of the chemotaxis signaling cascade. Fig 6A shows that the *cheA* mutant produced significantly less rot tissue compared to the WT SCRI1043, a phenotype that was complemented with a plasmid harboring the *cheA* gene (Fig 6A), indicating that chemotaxis is required to cause disease efficiently.

We then focused on the role of PacHIG chemoreceptors in virulence. Experiments with the quadruple mutant, Δ*pacHIG*, in which all four chemoreceptors genes of the cluster are deleted, showed a very significant reduction in the amount of rotten tissue produced compared to WT (Fig 6A), a phenotype that was complemented by providing the *pacG* gene *in trans* (S11 Fig). To identify which of these receptors were responsible for this virulence defect, strains with single deletions in *pacG*, *pacH* and *pacI* and a strain with a deletion of *pacC*, the chemoreceptor for aspartate, were tested for their ability to produce rot [29]. As shown in Fig 6B, the average rot weight recovered was significantly reduced in the *pacG* mutant, whereas the other mutants produced WT levels of rot tissue. Wild-type-like disease symptoms were observed when the *pacG* mutant was complemented with the pBBR*pacG* expression plasmid (S11 Fig). Previous studies from our laboratory have shown that chemoreceptors PacF, PacN and PacP mediate chemotaxis to formate [26], nitrate [27] and phosphorylated C3-compounds [30], respectively. Tuber slice infection assays with the three corresponding single mutants revealed no phenotypic alteration as compared to the WT (S12 Fig), a result that highlights the specificity of the effect observed with the *pacG* mutant.

*P. atrosepticum* infects not only the potato tuber but also the upper part of the plant causing blackleg disease [65]. We conducted assays where a wound was created in the leaf petiole, which was then covered with a drop of bacterial suspension. Bacteria that entered that tissue were quantified after 2 hours. Results obtained were analogous to those from the tuber-pathogenicity assay. A significant reduction in the number of *cheA* mutant bacteria recovered from inoculated plant tissue was observed compared to WT, and complementation *in trans* with *cheA* restored to WT levels. A statistically significant reduction was observed in the quadruple mutant as well as in the *pacG* mutant (Fig 6C). These results confirm a requirement of the PacG chemoreceptor for efficient virulence.

Several studies show that chemoreceptors possess functions in addition to mediating chemotaxis [66–68]. To assess this possibility, we conducted growth experiments in minimal and LB medium with the WT strain, the *pacG* mutant, and the mutant complemented with a plasmid harboring the *pacG* gene. In both media, the growth of the *pacG* mutant was slightly delayed (S13 Fig), and complementation of the mutant with the *pacG* gene resulted in WT growth. These data suggest that PacG signals to other regulatory circuits that affect growth.

Infection by *P. atrosepticum* requires the secretion of several enzymes that degrade the cell wall [65]. We tested protease, pectate lyase and cellulose activity in the WT, *pacG,* and complemented *pacG* mutant. We observed no difference in the production or secretion of these enzymes among the strains (S14 Fig).

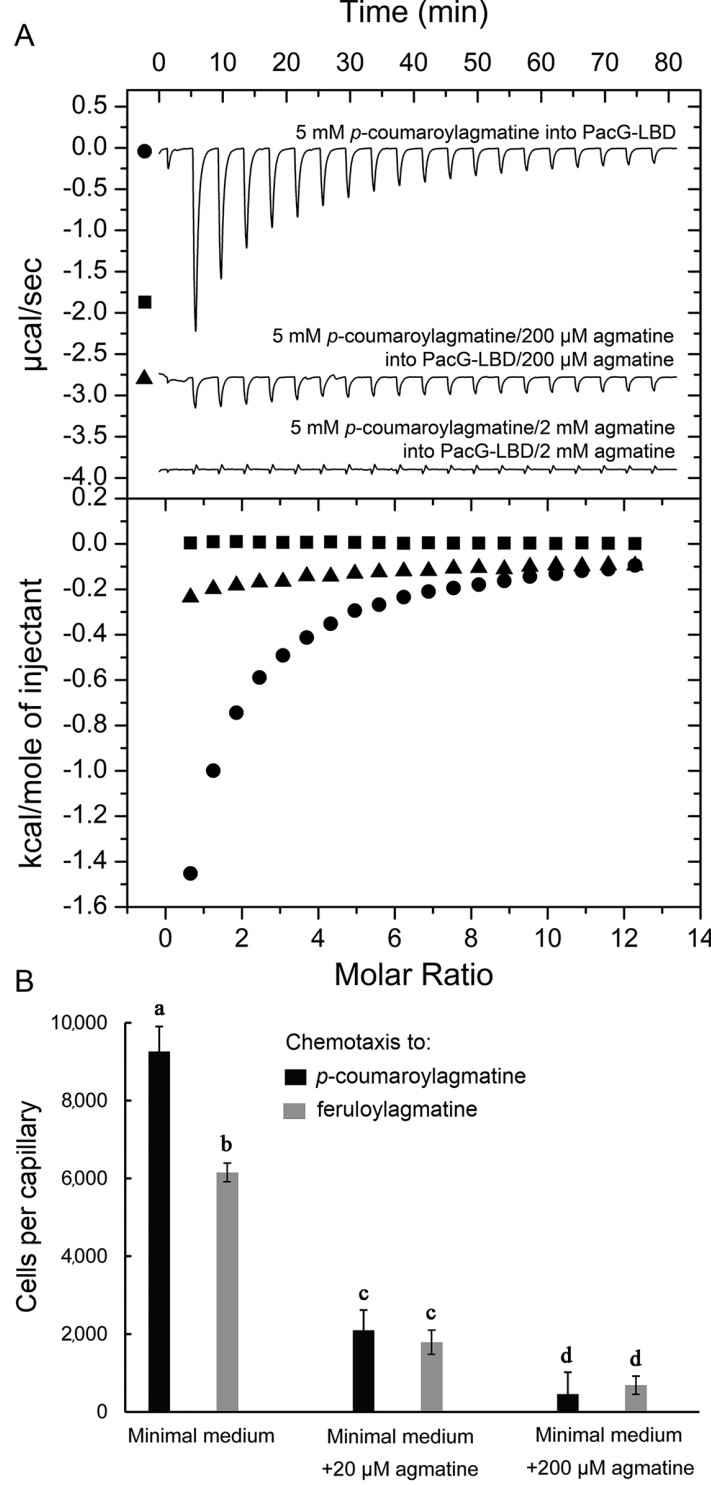

**Fig 5. Agmatine antagonizes the chemotaxis induced by *p*-coumaroylagmatine and feruloylagmatine. A**) Microcalorimetric competition assays *in vitro*: titration of 85 μM PacG-LBD with 5 mM *p*-coumaroylagmatine in the absence and presence of either 200 μM or 2 mM agmatine. Upper panels: raw titration data. Lower panels: concentration-normalized and dilution heat-corrected integrated raw data. **B**) Quantitative capillary assays of *P. atrosepticum* SCRI1043 to 50 μM *p*-coumaroylagmatine and feruloylagmatine in the presence of 20 or 200 μM agmatine. Data have been corrected for the number of cells that swam into buffer containing capillaries, namely 929 (minimal medium), 602 (minimal medium + 20 μM agmatine), 1,052 (minimal medium + 200

µM agmatine). The means and standard deviations from three biological replicates conducted in triplicate are shown. Unpaired t-test, different letters indicate that data are statistically different, p < 0.05.

### The three-dimensional structure of PacG-LBD

Because of its importance in virulence, we considered that it would be valuable to determine the structure of the PacG-LBD for comparison with the LBDs of other receptors in the Protein Data Bank. We reasoned that similar structures might identify other receptors that recognize ligands similar to those bound by PacG. To this end, we crystallized the LBD of PacG for determination of its 3D structure.

Two structures were solved to resolutions of 1.4 and 1.8 Å. The two structures contained either the monomer (pdb ID 9Q8B) or the physiologically relevant dimer (pdb ID 9Q8E) in the asymmetric unit (S5 Table). The protein shows the typical four-helix bundle fold (Fig 7A).

A structural alignment of the dimer with all proteins deposited in the Protein Data Bank showed that it was most similar to the LBD of the PcaY_PP chemoreceptor of *Pseudomonas putida*, which binds similar ligands derived from benzoate (S6 Table) [70]. Although PacG-LBD was co-crystallized with its ligands, no electron density corresponding to the ligands could be detected. Instead, densities corresponding to sulfate ions were seen in the presumptive ligand-binding sites (S15 Fig). It is likely that the presence of 2 M $(NH_4)_2SO_4$ in the crystallization buffer led to the replacement of the ligands by sulfate. Soaking the crystals with ligands did not displace the sulfate in the presumptive ligand-binding sites. Therefore, we conducted *in silico* molecular-docking experiments using the dimeric structure from which the sulfate and water molecules had been removed. Feruloylagmatine bound to the dimer interface at a site common to members of this receptor family [71,72] (Fig 7B,7C). Very similar docking was observed for *p*-coumaroylagmatine (S16 Fig).

To identify PacG homologs, we performed a BLAST search in the NCBI non-redundant protein sequence database, excluding the *Pectobacterium* taxa. The alignment of the most similar sequences is shown in S17 Fig. In this alignment, five of the amino acids in the presumed binding site (circled in Fig 7B) are fully conserved, suggesting that these homologs possess a similar ligand profile.

## Discussion

Our work here shows that one function of chemotaxis in phytopathogenic bacteria is chemotaxis to plant defense compounds. The combined action of three chemoreceptors may provide a mechanism by which the phytopathogen exploits the plant defense mechanisms to orient toward infected plants and infection sites. Notably, the identification of PacHIG homologs in other plant pathogens, including species of the *Burkholderia, Erwinia, Ralstonia, Pectobacterium* and *Dickeya* genera (S2 Table), suggests that this could be a general mechanism. None of the PacH and PacI ligands were of any apparent metabolic value (Fig 3B,3C), indicating that these compounds are cues that provide the bacterium with information on its environment. PacG ligands were of metabolic value (Fig 3B,3C), so these compounds may serve both as nutrient and signal molecules, as has been shown for other bacterial chemoattractants [73,74].

The existence of a single genetic locus with four chemoreceptor genes is atypical, since chemoreceptor genes are typically scattered over the genome [31]. Although their LBDs share only a modest sequence identity (S3 Table), there are significant parallels among their ligands: 1) The ligands are of similar structure since all are derivatives of benzoic and cinnamic acid (Fig 3A); 2) many of these ligands are synthesized and secreted following pathogen attack [34,35,45–47,75]; 3) their biosynthesis is interwoven (Figs 3A and 4) there is evidence that their signaling mechanisms are interconnected. For example, vanillin biosynthesis is induced in plant cells by salicylate [41]. Another example is the interference of benzoate with the degradation of *p*-hydroxybenzoate by *Xanthomonas campestris* [76]; the degradation of plant-produced *p*-hydroxybenzoate was found to be essential for *X. campestris* virulence [75,77]. Our study

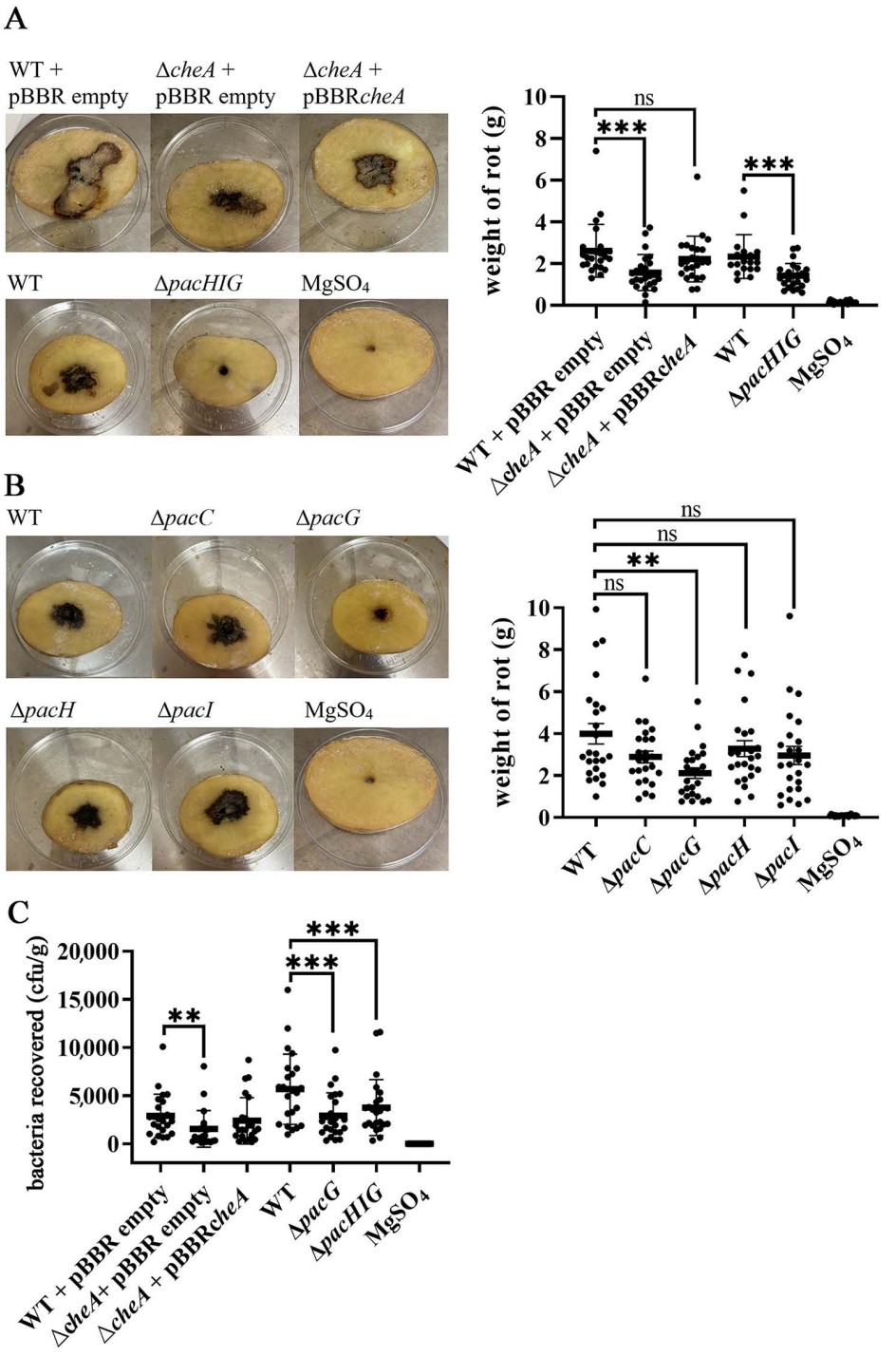

**Fig 6. Virulence assays of *P. atrosepticum* SCRI1043 strains.** Tuber-slice assays of WT and the Δ*cheA* strain containing the empty pBBR-based expression plasmid and the *cheA* strain harboring a pBBR-based *cheA* expression plasmid, and tuber slices inoculated with the WT, the quadruple mutant PacHIG, and MgSO₄ (control). Representative images of infected potato tubers are shown. **B)** Tuber-slice assay of the WT strain and strains deleted for *pacC* (chemoreceptor for aspartate), *pacG, pacH* and *pacI*. Representative images of infected potato tubers are shown. **C)** Petiole damage assays of WT and the Δ*cheA* strain containing the pBBR-based empty expression plasmid and *cheA* strain harboring a pBBR-based *cheA* expression plasmid, and petiole-damage assays using the WT, the quadruple mutant PacHIG, the mutant in *pacG*, and the MgSO₄ (control). The p-value was determined by the unpaired *t*-test: ** $p < 0.005$, *** $p < 0.0005$; ns: not significant.

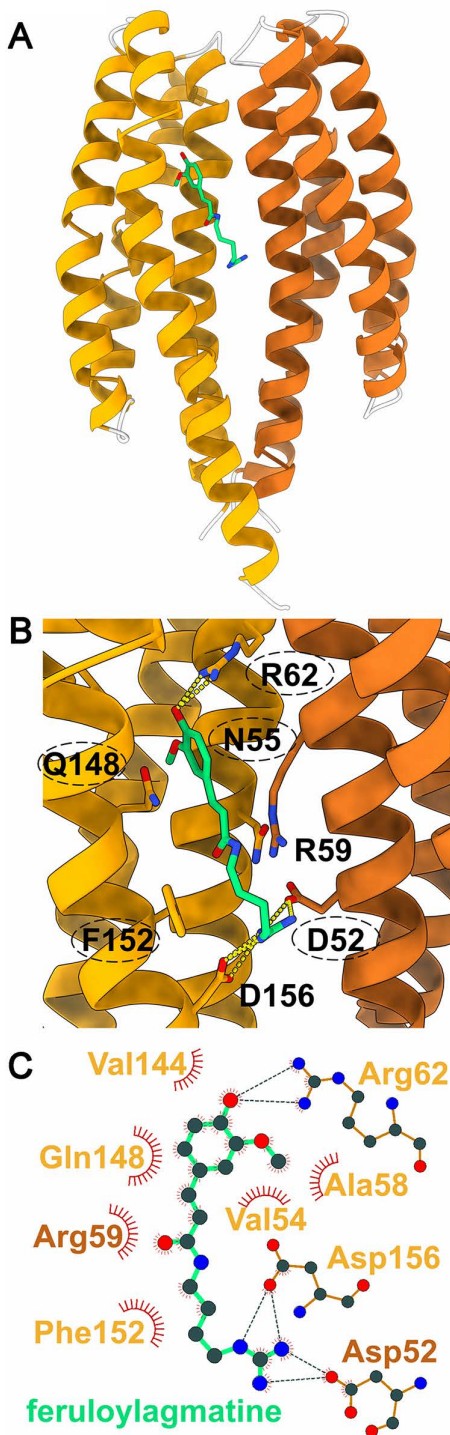

**Fig 7. The three-dimensional structure of PacG-LBD. A)** The three-dimensional structure of the PacG-LBD dimer at a resolution of 1.8 Å containing docked feruloylagmatine. The two chains of the dimer are colored differently. The structure with the highest confidence is presented. **B)** Zoom-in view of the ligand-binding pocket at the dimer interface. Amino acids involved in binding are shown in stick representation and are labelled. Dashed lines represent hydrogen bonds. Circled amino acids are fully conserved in a sequence alignment of homologous domains (S17 Fig). **C)** Schematic representation of ligand binding. The figure was produced using LigPlot [69].

may stimulate exploration of functional and evolutionary relationships among other chemoreceptors that are encoded within the same locus.

This is the first report of chemotaxis to feruloylagmatine and *p*-coumaroylagmatine [58]. Both compounds belong to the family of hydroxycinnamic acid amides that are well-known plant defense metabolites [48]. An accumulation of *p*-coumaroylagmatine and feruloylagmatine has previously been reported following bacterial or fungal infection of different plants, including wheat [50,78], *Arabidopsis* [79,80], rice [52], potato [51], and purple false brome [81]. Moreover, in tobacco plants infected with *P. atrosepticum*, genes for the conversion of arginine to agmatine, and of agmatine to *p*-coumaroylagmatine, were upregulated [82]. Jasmonic acid is a central plant signal molecule that coordinates plant defenses [83] and increases the expression of proteins required for feruloylagmatine and *p*-coumaroylagmatine synthesis and secretion, resulting in an increased resistance against necrotrophic fungal pathogens [84,85]. Increases in feruloylagmatine were also observed in response to other stresses such as wounding [52] or cold [86]. The plant defense properties of these compounds is generally attributed to their antibacterial and antifungal activity [81,87]. The protective activity of these compounds was evidenced by the severe virulence of a fungal pathogen in an *Arabidopsis thaliana* mutant plant deficient in *p*-coumaroylagmatine synthesis [88] and the increased resistance to fungal attack of *A. thaliana* [89] and *Torenia hybrida* plants that accumulate large amounts of *p*-coumaroylagmatine [53].

We report here a ligand that acts as an antagonist. In contrast to feruloylagmatine and *p*-coumaroylagmatine, agmatine binding to PacG-LBD did not trigger chemotaxis. We show that agmatine competes *in vitro* with *p*-coumaroylagmatine for binding at PacG-LBD (Fig 5A). Chemotaxis assays showed that the presence of agmatine reduced the magnitude of chemotaxis to feruloylagmatine and *p*-coumaroylagmatine (Fig 5B). The presence of such signal antagonists appears to be a common phenomenon in signal transduction. It has been observed in chemoreceptors with different classes of LBDs, including four-helix bundle [90], sCache [63] and dCache [64] domains. A similar behavior has also been observed for sensor histidine kinases [59,91] and transcriptional regulators [61,92]. It is currently unclear whether the existence of naturally occurring antagonists is of physiological relevance or represents imperfections in the evolution of molecular recognition. Potentially, the action of antagonists may result in a fine-tuning of responses.

PacH and PacI bind the plant hormone salicylate and some of its derivatives. Salicylate is a key plant hormone required for establishing resistance against many pathogens [93]. Like the agmatine derivatives, phytopathogen infection increases production of salicylate by the plant, inducing the expression of plant defense genes that provide systemic acquired resistance [93–96]. *p*-hydroxybenzoate is another important phenolic compound involved in plant defense responses against pathogen attack. Its biosynthesis is promoted by bacterial and fungal pathogens [46,75]. Furthermore, the composition of root exudates can be altered by stress, resulting in an increased secretion of salicylate in plants grown under low phosphate conditions [97]. Elevated temperatures and extended photoperiods have been observed to enhance the levels of benzoic and *p*-hydroxybenzoate in cucumber [98]. Based on these observations, the authors of that study proposed that benzoate serves as a plant defense compound. Plant-produced *p*-methoxybenzoate was also found to inhibit production of virulence factors in *P. aeruginosa* [47].

However, how do the concentrations of plant defense compounds detected in plants following pathogen attack correspond to the sensitivity of the corresponding chemotactic responses? A number of studies have quantified levels in infected and control plants. In one study cucumber plants were infected with *Pseudomonas syringae* and the concentration of salicylate and *p*-hydroxybenzoate quantified after 20 hours [99]. Whereas both compounds were either undetectable or at a concentration close to the detection threshold in uninfected control plants, salicylate and *p*-hydroxybenzoate concentrations rose to up to about 300 or 100 µM, respectively. The results of this study are very similar to an analogous study using a different model system, namely the infection of tobacco plants with the Tobaco mosaic virus [100]. In analogy to the above study, salicylate was undetectable in mock-inoculated leafs. However, 40 hours post-inoculation, a time before lesions become visible, the local salicylate concentration increased to up to 380 µM. The same study also showed that *P. syringae* infected tobacco plants also had largely increased salicylate levels. These concentrations are well above

the onset concentrations for salicylate and *p*-hydroxybenzoate of 0.5 μM and 1 μM, respectively (Fig 4). Another study has investigated *p*-coumaroylagmatine levels in *Arabidopsis thaliana* infected with *Alternaria brassicola* [88]. The authors show that this compound accumulated at concentrations of up to about 35 μM in infected plants, whereas its concentration in control plants was more than 10 times lower. This concentration is close to the *p*-coumaroylagmatine concentration that induced maximal chemotaxis (Fig 4).

The physiological relevance of salicylate is furthermore reflected in the fact that a significant number of salicylate chemoreceptors have been identified. These belong to two different superfamilies, namely those with dCache [101] and four-helix bundle LBDs [70,102], indicative of convergent evolutionary processes. Sequence alignments of the LBDs of PacH and PacI with other salicylate-binding four-helix bundle LBDs reveal identities between 13–18% (S3 Table, S18 Fig), indicating that chemoreceptor function is not reflected in similarity of the LBD sequence.

The deletion of chemotactic signaling genes in many human and plant pathogens reduces host colonization and virulence [19,20], indicating that chemotactic ability is a virulence factor. Our results corroborate these studies as the deletion of the central histidine kinase *cheA* gene reduced *P. atrosepticum* virulence (Fig 7A). However, little is known about the chemoreceptors and their signals that are relevant for virulence. We have shown that strains with deletions of the *pacH*, *pacI*, *pacC*, *pacF*, *pacN* and *pacP* chemoreceptor genes showed normal virulence (Figs 7, S12). However, deletion of *pacG* reduced virulence both in the potato tuber and petiole damage assays (Fig 7). This defect was not caused by any decrease in secretion of enzymes that degrade the cell wall (S14 Fig), which is one of the main virulence factors of *P. atrosepticum* [65]. However, growth of this mutant was also delayed (S13 Fig). We conclude that PacG affects growth as well as chemotaxis, and it is therefore uncertain whether the virulence defect is caused by decreased chemotaxis or slower growth.

There is increasing evidence that chemoreceptors communicate with signaling circuits other than chemotaxis. For example, mutation in chemoreceptors for nitrate, amino acids and GABA in the phytopathogen *P. syringae* had several additional effects in addition to chemotaxis, including alterations in gene expression, biofilm formation, c-di-GMP levels and swarming behavior [10,66,67]. In *Comamonas testosteroni,* the chemotaxis pathway was found to cross-talk with a pathway that regulates biofilm formation [68].

In summary, this study identifies a dedicated cluster of four chemoreceptor genes in a global phytopathogen. The products of at least three of these genes (*pacHIG*) mediate chemotaxis toward plant defense compounds, and the PacG receptor contributes significantly to virulence. The ability to migrate toward the source of these compounds suggests that chemotaxis brings bacteria to sites of wounding to facilitate infection at such sites.

## Materials and Methods

### Strains, plasmids, and culture conditions

The bacterial strains, plasmids, and oligonucleotides used are listed in S7 and S8 Tables. Unless otherwise stated, *P. atrosepticum* strains were cultured at 28 °C in LB or minimal medium (MM) (0.41 mM $MgSO_4$, 7.56 mM $(NH_4)_2SO_4$, 40 mM $K_2HPO_4$, 15 mM $KH_2PO_4$) supplemented with 20 mM glucose as carbon source. *E. coli* strains were grown in LB medium at 37 °C. *E. coli* DH5α was used as a host for gene cloning. Media for propagation of *E. coli* β2163 were supplemented with 300 μM 2,6-diaminopimelic acid. When necessary, antibiotics were used at the following concentrations (μg/mL): streptomycin, 50; kanamycin, 50; ampicillin, 100; gentamicin, 10 (*E. coli*) or 25 (*P. atrosepticum*).

### Overexpression and purification of proteins

*E. coli* BL21(DE3) harboring plasmids pET28-ECA_RS21440-LBD, pET28-ECA_RS21445-LBD, pET28-ECA_RS21450-LBD and pET28-ECA_RS21455-LBD were grown in LB medium, supplemented with kanamycin, until an $OD_{660}$ of 0.5, at which protein expression was induced by the addition of 0.1 mM isopropyl-β-D-thiogalactopyranoside (IPTG). Growth was

continued overnight at 18 °C, prior to cell harvest by centrifugation at 20,000 x $g$ for 20 min. Cells were then broken by French press treatment at a gauge pressure of 62.5 lb/in$^2$, and proteins were purified by metal affinity chromatography following the protocol reported in [28]. PacH-LBD and PacG-LBD were dialyzed for immediate analysis into 5 mM Tris, 5 mM PIPES, 5 mM MES, 10% (v/v) glycerol, pH 7.0, whereas PacI-LBD was dialyzed into this buffer at pH 6.0. The sequences of proteins used in this study are provided in S9 Table.

### Thermal shift assay

Differential scanning fluorimetry assays were performed using a BioRad MyiQ2 Real-Time PCR instrument, as previously described (15). Compound arrays PM1, PM2A, PM3B, PM4A and PM5 (Biolog, Hayward CA, USA), containing C-, N-, P-, S- sources and nutrient supplements, were used (S3 Fig). Ligand solutions were prepared by dissolving the array compounds in 50 μL of Milli-Q water, which, according to the information provided by the manufacturer, corresponds to a concentration of 10–20 mM.

### Isothermal titration calorimetry

Measurements were made using a VP-ITC microcalorimeter (MicroCal, Inc., Northampton, MA) at 25 °C. Proteins at 32–100 μM were placed into the sample cell and titrated with 6.4 to 14.4 μL aliquots of 1–5 mM ligand solutions made up in dialysis buffer. Data were analyzed with SEDPHAT [103] using a model for two-site interaction of the type A+B+B<-->AB+B<-->ABB, with 2 symmetric sites, and plotted by GUSSI.

### Generation of chemoreceptor mutants

Mutants in chemoreceptors genes *ECA_RS21440* (*pacH*), *ECA_RS21445* (*pacI*) and *ECA_RS21455* (*pacG*), as well as the quadruple mutant strain Δ*pacH-pacI-ECA_RS24450-pacG* (Δ*pacHIG),* were generated by homologous recombination using derivative plasmids of the suicide vector pKNG101 (S7 Table). Briefly, the up- and downstream flanking regions of each target gene were PCR-amplified using the primers detailed in S8 Table. The resulting plasmids pKNG101_ΔECA_ RS21440, pKNG101_ΔECA_RS21445, pKNG101_ΔECA_RS21455, and pKNG101_PacHIG were transferred to *P. atrosepticum* SCRI1043 by biparental conjugation using *E. coli* β2163 as the donor strain. All plasmids and the resultant Δ*ECA_RS21440* (Δ*pacH),* Δ*ECA_RS21445* (Δ*pacI),* Δ*ECA_RS21455* (Δ*pacG*) and PacHIG mutant strains were verified by PCR and DNA sequencing.

### Chemotaxis assays

Overnight cultures of *P. atrosepticum* SCRI1043 strains in minimal medium supplemented with 20 mM glucose as a carbon source were used to inoculate fresh minimal medium to an initial $OD_{660}$ of 0.1. To test chemotaxis to the compounds recognized by PacI and PacH, benzoate was added to the minimal medium to a final concentration of 500 μM 40 minutes after inoculation [in analogy to a study of benzoate chemotaxis in *P. putida* [104]]. Cells were then cultured at 28 °C until an $OD_{660}$ of 0.4–0.5. Subsequently, cells were washed twice by centrifugation and resuspension in chemotaxis buffer (50 mM potassium phosphate, 20 μM EDTA, 0.05% (v/v) glycerol, pH 7.0) and then resuspended in the same buffer to reach an $OD_{660}$ of 0.1. Aliquots of 230 μL of the resulting cell suspensions were placed into wells of 96-well microtiter plates. One-μL capillaries (Microcaps, Drummond Scientific, Broomall, PA, USA) were heat-sealed at one end and filled with chemotaxis buffer (control) or chemoeffector solutions prepared in the same buffer. Subsequently, the capillaries were washed with sterile water, immersed into the bacterial suspensions at their open end, and incubated for 30 min at room temperature. Capillaries were then removed, rinsed with sterile water, and emptied into 1 mL of chemotaxis buffer. Serial dilutions were plated on minimal medium plates, prior to an incubation at 30 °C for 48 h. Colonies were counted, and the data were corrected with the number of cells that swam into chemotaxis buffer containing capillaries. For the chemotaxis

assay of feruloylagmatine and *p*-coumaroylagmatine, polybuffer (5 mM Tris, 5 mM PIPES, 5 mM MES, 10% (v/v) glycerol, pH 7.0) was used instead of chemotaxis buffer described above. Data are means and standard deviations of at least three biological replicates conducted in triplicate.

## Synthesis and analysis of *p*-coumaroylagmatine and feruloylagmatine

*p*-Coumaroyl- and feruloyl-*N*-hydroxysuccinimide esters were synthesized from *p*-coumaric acid and ferulic acid, respectively, according to [105]. *p*-Coumaroylagmatine and feruloylagmatine were synthesized from the corresponding *N*-hydroxysuccinimide esters and agmatine sulfate (Sigma Aldrich, St Louis, MO, USA) following the protocol reported by [106]. The authenticity of the compounds was confirmed by comparing their $^1$H-NMR spectra (S6 Fig) with previously reported spectra [107,108]. The $^1$H-NMR spectra were recorded on a Bruker AVANCE 400 spectrometer (Bruker, Billerica, MA, USA) in $CD_3OD$.

## Determination of minimal inhibitory concentrations (MIC)

Experiments were conducted in 96-well micro test plates (Sarstedt AG & Co, KG., Nümbrecht, Germany). Each well was filled with 200 µL of minimal medium supplemented with 20 mM glucose as a carbon source. Wells were inoculated with overnight cultures grown in the same medium to an $OD_{660}$ of 0.02. Test compounds were added individually at concentrations ranging from 10 µM to 50 mM. The MIC was defined as the lowest compound concentration that prevented growth after 48 h at 28 °C.

## Growth experiments

*P. atrosepticum* was cultured overnight in minimal medium supplemented with 20 mM glucose as a carbon source. Cultures were washed and then diluted to an $OD_{660}$ of 0.02 in the same medium supplemented with different carbon sources at a final concentration of 5 mM. Subsequently, 200 µL aliquots were transferred into microwell plates, and growth at 28 °C was followed on a Bioscreen microbiological growth analyzer for 48 h (Oy Growth Curves Ab Ltd., Helsinki, Finland).

## RNA extraction, cDNA synthesis and PCR analyses

*P. atrosepticum* cultures grown overnight in minimal medium supplemented with 20 mM glucose as a carbon source were used to inoculate fresh medium to an $OD_{660}$ of 0.1. At an $OD_{660}$ of 0.5, 14 mL samples were collected by centrifugation at 8,000 x *g* at 4 ºC for 5 minutes. RNA extraction and cDNA synthesis were performed as described in [109]. Briefly, total RNA was extracted using the TRI Reagent (Invitrogen), followed by Turbo DNase treatment (Ambion) and RNA clean-up using the RNeasy Mini Kit (Qiagen), following the manufacturer's instructions. RNA concentration was determined spectrophotometrically and RNA integrity was assessed by electrophoresis on 2% (w/v) agarose gels. cDNA synthesis was performed using random hexamers (GE Healthcare) and SuperScript II reverse transcriptase (Invitrogen) in a reaction volume of 20 µL containing 1.5 µg of total RNA. Samples were incubated at 42 °C for 2 h. As negative control the reaction was performed omitting the reverse transcriptase. For the RT-PCR analysis, the equivalent of 50 ng of total RNA was subjected to PCR amplification using primers to amplify across the junctions (S8 Table). Positive and negative control PCR reactions were performed using genomic DNA and no-RT cDNA samples, respectively, as templates. Oligonucleotides amplifying the constitutive gene *gyrB* were also used as internal control.

## Virulence assays: Tuber slice assay

The assays was performed as previously described in [110]. Briefly, strains were grown overnight in LB media at 28 ºC, washed twice with 10 mM $MgSO_4$, and resuspended in the same solution to a final concentration of $10^7$ CFU/mL. Potato tubers (cv. Maris Piper) were surface-sterilized with 1% (w/v) Virkon for 10 minutes, rinsed with sterile distilled water, and

sliced into 1 cm-thick sections using an ethanol-sterilized knife. Wells (5 mm deep, 7 mm diameter) were made at the center of each slice and inoculated with 5 x $10^5$ CFU in 50 µL or 50 µL of 10 mM $MgSO_4$ (control). For each condition, 6 slices from different tubers were used. Slices were placed randomly on moist tissue in Petri dishes and incubated in a sealed container at 20 ºC for 5 days. Disease severity was quantified by removing and weighing the macerated tissue. Each experiment was repeated at least three times with independent cultures. Data were analyzed and plotted using GraphPad Prism v10.1.2.

## Petiole damage assay

Potato plants (cv. Estima) were grown in compost for 4 weeks to a height of about 20 cm. The petiole epidermal surface was lightly scraped 5 times with a syringe needle. Overnight cultures of *P. atrosepticum* strains grown in LB broth at 28 ºC were washed twice with 10 mM $MgSO_4$ and resuspended in the same solution to a final concentration of $10^7$ CFU/mL. Using a micropipette tip, 10 µL drops of the bacterial suspension or 10 mM $MgSO_4$ (control) were applied to the wounds. After 2 h, the plants were sprayed with 70% (v/v) ethanol to sterilize the surface, and 5 cm stem segments adjacent to the infected petiole were excised. Samples were placed in universal long extraction bags (Bioreba, Reinach, Switzerland) containing 10 mL of 0.25x Ringer´s solution (2.25 g/L NaCl, 0.105 g/L KCl, 0.12 g/L $CaCl_2.2H_2O$, 0.05 g/L $NaHCO_3$) and homogenized with a Homex 6 (Bioreba) instrument. Homogenates were serially diluted in 0.25x Ringer´s solution and plated on semi-selective crystal violet pectate medium [111]. CFUs were counted after incubation at 28 ºC for 48 h. Three biological replicates were made, each containing 8 different plants.

## Protein crystallization, structure resolution and molecular docking

Purified PacG-LBD in 3 mM Tris/HCl, 3 mM Pipes, 3 mM MES was adjusted to pH 8.0, and supplemented with 10% (v/v) glycerol. The protein was then concentrated to ~10 mg/mL using 10 kDa molecular weight cut-off Centricon concentrators (Amicon) at 4 ºC. In parallel, PacG-LBD was incubated with agmatine or *p*-coumaroylagmatine at 4 °C. The excess of ligands was removed by multiple rounds of buffer exchange during the concentration step. Sitting-drop vapor diffusion crystallization trials were made in 96-well MRC 2-Drop plates (Molecular Dimensions). Drops, consisting of 1 µL of protein solution and 1 µL of reservoir solution, were equilibrated against 50 µL of reservoir solution. Commercially available crystallization screens HR 1 and II (Hampton Research Crystal Screens I and II) were used. Plates were incubated at 20 °C. Crystals were transferred to a drop of mother solution supplemented with 15% (v/v) glycerol as cryo-protectant and immediately flash-cooled in liquid nitrogen for storage. Soaking experiments were done by including 1 mM of ligand (in DMSO) to the cryo-protectant solution. Crystals diffracted at beamline ID23–2 of the European Synchrotron Radiation Facility (ESRF, Grenoble, France), and beamline XALOC at the ALBA Synchrotron Light Source (Barcelona, Spain). Crystals grown 0.2 M $(NH_4)_2SO_4$, 0.1 M Na acetate trihydrate, pH 4.6, 30% (w/v) PEG 2K and 0.2 M K Na tartrate tetrahydrate, 0.1 M Na citrate dehydrate, pH 5.6, 2.0 M $(NH_4)_2SO_4$ diffracted to a resolution below 2.0 Å. Diffraction data were processed using XDS for indexing and integration [112], and scaled using AIMLESS within the CCP4 software suite [113]. Phases were obtained using Arcimboldo_Lite [114] by searching for two thirty amino acid helices with the orthorhombic data set at 1.40 Å resolution. It located 135 alanine residues used for initial refinement with Refmac [115]. Model building and refinement followed an iterative approach, combining automatic refinement using REFMAC5 [115] and PHENIX.REFINE [116] with manual model correction and water selection in COOT [117]. Torsion-Libration-Screw parameterization [118] was applied in the final stages of refinement to improve atomic displacement modeling. Model validation was performed using MOLPROBITY [119], and through the PDB validation server prior to deposition in the Protein Data Bank. A summary of crystallographic data collection, processing, and model refinement statistics is provided in S5 Table. Coordinates and structure factors have been deposited at the protein data bank with ID 9Q8B and 9Q8E. For the *in silico* docking, ligands were downloaded from PubChem database in SDF format. The PacG-LBD crystal structure of (pdb ID 9Q8E) was cleaned by removing co-crystallized sulfate and water molecules. DiffDock was used to computationally dock the ligands using the

default settings [120]. The docked structures were subjected to short steepest descent minimization followed by simulated annealing minimization using AMBER14 force field: ff14SB [121] until convergence was reached to remove any conformational stress in the system as described in [122]. Resulting models were viewed and interpreted using ChimeraX 1.9.

### Exoenzyme assays: Protease assays

The protease assay was adapted from [123]. Briefly, *P. atrosepticum* strains were grown in LB broth at 28 ºC overnight, cells were then washed twice with chemotaxis buffer and resuspended in the same solution to a final $OD_{660}$ of 1. Aliquots (10 µL) were spotted onto protease assay plates (LB broth containing 1% (w/v) skimmed milk) that were incubated at 28 ºC during 72 h, and then inspected visually.

### Pectate lyase and cellulase agar plate assays

The pectate lyase and cellulase assays were adapted from [124] and [125], respectively. Aliquots (5 µL) of overnight *P. atrosepticum* cultures (adjusted to an $OD_{660}$ of 1.0 by dilution with LB medium) were spotted onto pectate lyase assay plates [1.6% (w/v) bacto agar, 0.1% (w/v) yeast extract, 0.1% (w/v) $(NH_4)_2SO_4$, 1 mM $MgSO_4$, 0.5% (v/v) glycerol, 0.5% (w/v) polygalacturonic acid (sodium salt) in 20% (v/v) phosphate buffer (15 g/L $Na_2HPO_4$, 0.7 g/L $NaH_2PO_4.H_2O$, pH 8.0)] or cellulase assay plates [1.6% (w/v) bacto agar, 1% (w/v) carboxymethylcellulose, 0.5% (w/v) yeast extract, 0.2% (v/v) glycerol, 2% (v/v) 50× phosphate buffer (350 g/L $K_2HPO_4$, 100 g/L $KH_2PO_4$, pH 6.9–7.1), 0.1% (w/v) $(NH_4)_2SO_4$, 0.01% (w/v) $MgSO_4$]. Exoenzyme production was visualized after an incubation at 28 °C for 5 days. For pectate lyase activity, plates were overlaid with 7.5% (w/v) copper acetate for 1 h and enzyme activity was visualized as double cream-colored haloes on a translucent blue-green background. For cellulase activity, plates were sequentially stained with 0.2% (w/v) congo red for 20 min, bleached with 1 M NaCl for 15 min, and then stained with 1 M HCl for 5 min. Enzymatic zones correspond to orange-red haloes on a dark blue background.

### Supporting information

**S1 Fig. Transcript analysis by RT-PCR using primers designed to span the intergenic region between two adjacent genes.** A) Schematic view of the locus encoding the four chemoreceptor genes under investigation. The red brackets indicate the position of primer pairs for the analyses shown below. B) For each region, three PCR analyses were carried out: C: positive control with genomic DNA as template; -: negative control with no reverse transcriptase; + : RT-PCR on cDNA. Oligonucleotides amplifying the constitutive gene *gyrB* were used as a positive control for the reverse transcription-polymerase chain reaction. Samples for RNA isolation were taken at mid-logarithmic phase like the cells used for the chemotaxis assays.
(DOCX)

**S2 Fig. RT-PCR analysis of *pacH, pacI and pacG* gene transcripts in *Pectobacterium atrosepticum* SCRI1043.** PCR analyses were performed using internal primers pairs for each gene: C, positive control with genomic DNA as template; + , RT-PCR on cDNA; -, negative control with no reverse transcriptase. The constitutively expressed gene *gyrB* was used as a positive control. The following primer pairs were used: RS21440-FW-RT-PCR/RS21440-RV-RT-PCR, RS21445-FW-RT-PCR/RS21445-RV-RT-PCR, RS21455-NdeI-F/RS21455-PstI-R and ECA-gyrB-F-qPCR/ECA-gyrB-R-qPCR (S8 Table). Samples for RNA isolation were taken at mid-logarithmic phase, the same experimental conditions used for the chemotaxis assays, and correspond to the same samples used in Fig 1B.
(DOCX)

**S3 Fig. The composition of Biolog compound arrays PM1, PM2A, PM3B, PM4A and PM5.**
(DOCX)

**S4 Fig. Microcalorimetric titration of ECA_RS21440-LBD (PacH-LBD) with salicylate and vanillin.** A) Titration of 44 µM PacH-LBD with 12.8 µL aliquots of 5 mM salicylate. B) Titration of 63 µM protein with 12.8 µL aliquots of 5 mM vanillin. Upper panels: Raw titration data. Lower panels: Concentration-normalized and dilution heat-corrected integrated raw data. The derived dissociation constants are provided in Table 1.
(DOCX)

**S5 Fig. Microcalorimetric binding studies of capric acid to PacI-LBD.** Shown are the raw data for the titration of 50 µM PacI-LBD with 12.8 µl aliquots of 1 mM capric acid. Heat changes are small and uniform and comparable to the titration of buffer with capric acid, indicative of an absence of binding. The scale on the y-axis corresponds to that of Fig 2B.
(DOCX)

**S6 Fig. $^1$H-NMR spectra of *p*-coumaroylagmatine and feruloylagmatine (400 MHz, CD$_3$OD).** Proton signal assignments are indicated in the spectra and correspond to the numbering shown in the chemical structures.
(DOCX)

**S7 Fig. The chemotaxis to vanillin (A), salicylate (B) and benzoate (C) is induced by benzoate.** Quantitative chemotaxis capillary assays of *P. atrosepticum* SCRI1043 to different concentrations of vanillin, salicylate and benzoate when grown in minimal medium (MM) or MM supplemented with 500 µM of vanillin or benzoate. Data have been corrected with the number of bacteria that swam into buffer containing capillaries namely 1,537 (vanillin in MM), 50 (vanillin in MM+vanillin), 2,771 (vanillin in MM+benzoate), 1,537 (salicylate in MM), 2,324 (salicylate in MM+benzoate), 720 (benzoate in MM), and 986 (benzoate in MM+benzoate).
(DOCX)

**S8 Fig. Quantitative capillary chemotaxis assays of *P. atrosepticum* SCRI1043 to 5 mM L-Asp for cells grown in minimal medium and minimal medium containing 500 µM benzoate.** Data have been corrected with the number of bacteria that swam into buffer-containing capillaries, namely 682 (in minimal medium) and 1,197 (in minimal medium+500 µM benzoate).
(DOCX)

**S9 Fig. Quantitative chemotaxis capillary assays of *P. atrosepticum* SCRI1043 strains to 5 mM L-Asp (A) and 0.1% (w/v) casamino acids (B).** Strains for experiments shown in panel A were grown in minimal medium and 500 µM benzoate was added 40 minutes after inoculation. Strains for experiments shown in panel B were grown in minimal medium. Data have been corrected with the number of bacteria that swam into buffer containing capillaries, namely 1,197 (wild type, panel A), 1,777 (ΔpacH), 1,675 (ΔpacI), 963 (wild type, panel B) and 918 (ΔpacG).
(DOCX)

**S10 Fig. Chemotaxis to agmatine in under aerobic (A) and anaerobic (B) growth conditions.** A) Quantitative capillary assays of *P. atrosepticum* SCRI1043 grown aerobically in minimal medium with glucose or agmatine as carbon source or in potato broth. The agmatine concentrations are indicated. Data have been corrected for the number of cells that swam into buffer-only capillaries: 1,685 (glucose as carbon source), 1,462 (agmatine as carbon source) and 1,293 (potato broth). B) Quantitative capillary assays of *P. atrosepticum* SCRI1043 grown anaerobically in minimal medium with glucose as carbon source. The agmatine concentrations are indicated. Data have been corrected for the number of cells that swam into buffer-only capillaries (690). The means and standard deviations from three biological replicates conducted in triplicate are shown.
(DOCX)

**S11 Fig. Complementation experiments of plant infection assays.** Tuber-slice assays of *P. atrosepticum* SCRI1043 strains. Wild type (WT) or *pacG* and *pacHIG* mutants harbouring either the empty expression plasmid pBBR empty or the

pBBR*pacG* expression plasmid were used to infect potato tubers. Inoculation with a MgSO$_4$ solution served as control. Representative images of the phenotypes observed are shown in the upper part. The statistical analysis of one of three experiments is shown in the lower part. *** $p < 0.0005$ in unpaired *t*-test.
(DOCX)

**S12 Fig. Soft root tissue production in potato inoculated with different *P. atrosepticum* SCRI1043 strains.** Results from tuber slice assays. Chemoreceptors PacF, PacN and PacP were shown to mediate chemotaxis to formate (1), nitrate (2) and phosphorylated compounds (3), respectively. MgSO$_4$ was added as a negative control. The p-value was determined by the unpaired *t*-test: **** $p < 0.0001$; ns: not significant.
(DOCX)

**S13 Fig. Growth experiments of *P. atrosepticum* SCRI1043, a mutant deficient in the *pacG* gene, and the mutant strain complemented with a plasmid harboring the *pacG* gene in minimal medium supplemented with 20 mM glucose as sole C-source (A) and LB medium (B).** Data are means and standard deviations from three biological replicates.
(DOCX)

**S14 Fig. Assessment of the impact of the deletion of *pacG* on the production of plant cell wall–degrading enzymes.** Experiments were conducted with the wild type (WT) strain, the mutant in the *pacG* gene (Δ*pacG*) and the mutant complemented with a plasmid harboring the *pacG* gene (compl). A) Skimmed milk agar plate assay of protease production. B) Pectate lyase assay. C) Cellulase assay.
(DOCX)

**S15 Fig. Three-dimensional structure of PacG-LBD.** Although the protein was co-crystallized with agmatine and *p*-coumaroylagmatine, no electron density accounting for these compounds was visualized. Instead, the protein dimer contained four sulfate molecules, of which two in the binding pocket. Sulfate molecules are shown in stick mode. The |2Fo-Fc| map is contoured at 2.0 σ (in blue). The protein was crystallized in the presence of 2 M (NH$_4$)$_2$SO$_4$.
(DOCX)

**S16 Fig. Molecular docking of feruloylagmatine (left) and *p*-coumaroylagmatine (right) to the structure of PacG-LBD.** The monomers of the dimer are colored differently.
(DOCX)

**S17 Fig. Alignment of the PacG-LBD sequence with those of homologous domains.** A protein-protein BLAST (blastp) (4) search in the NCBI non-redundant protein sequence database was conducted using the PacG-LBD sequence as query. Default parameters were used, and sequences of the taxid *Pectobacterium* (taxid:122277) were excluded. The alignment was done using the CLUSTALW algorithm of the NPS@ software (5). The Gonnet protein weight matrix was used; gap opening and gap extension penalties were 10.0 and 0.1, respectively. Residues in red are identical, green highly similar and blue weakly similar. Five amino acids present in the ligand binding pocket are fully conserved and shaded in yellow. The corresponding amino acids are circled in Fig 7B.
(DOCX)

**S18 Fig. Sequence alignment of four-helix bundle LBDs of chemoreceptors that bind salicylate.** Pairwise sequence alignments of the LBDs of PacI and PacH with those of PcaY_PP (6) and PcpI (7). The alignment was done using the CLUSTALW algorithm of the NPS@ software (5). The Gonnet protein weight matrix was used; gap opening and gap extension penalties were 10.0 and 0.1, respectively. Residues in red are identical, green highly similar and blue weakly similar.
(DOCX)

**S1 Table. Degree of plant specificity (DPS) scores of the *P. atrosepticum* SCRI1043 chemoreceptors.** Data were extracted from (8). The authors clustered all available chemoreceptor LBD sequences. They then quantified the relative amount of chemoreceptors from plant-associated bacteria (PAB) and plant pathogens (PP) in each of the LBD clusters. The DPS score corresponds to the abundance of receptors from PAB and PP in each of the clusters. A score of 100 indicates that all sequences of a cluster are from PAB/PP and a score of 0 indicates none. Names and ligands of characterized chemoreceptors are also indicated. The chemoreceptors studied in this article are in bold.
(DOCX)

**S2 Table. Composition of the LBD clusters that contain chemoreceptors PacH, PacI and PacG.** It is indicated whether the corresponding strain is a plant-associated bacterium (PAB) or a phytopathogen (PP). Data are taken from (8).
(DOCX)

**S3 Table. Amino acid sequence identity (%) of pairwise alignments of the LBDs from different chemoreceptors.** A) Sequence identities of 4 chemoreceptors from the gene cluster of *P. atrosepticum* SCRI1043 studied in this work. B) Sequence identities between the four-helix bundle type LBDs of chemoreceptors that bind salicylate. The LBDs of the PcaY-PP and PcpI chemoreceptors from *Pseudomonas putida* KT2440 and *P. putida* 1290 bound salicylate (6, 7). Alignments were made with the BLAST tool of NCBI. The LBD was defined as the protein segment between the two transmembrane regions as predicted by TMHMM – 2.0 (11).
(DOCX)

**S4 Table. List of compounds that failed to bind in microcalorimetric experiments to the LBDs of chemoreceptors PacH, PacI and PacG.** Microcalorimetric titrations were conducted with the highest possible ligand concentrations, i.e., that induced dilution heats inferior to 0.1 µcal/sec when injected into buffer. These concentrations were in the range of 1–5 mM. Listed are the compounds that did not produce binding heats when injected into 32–100 µM protein solutions.
(DOCX)

**S5 Table. Resolution of the three-dimensional structure of PacG-LBD by X-ray crystallography: Data collection and refinement statistics.** Statistics for the highest-resolution shell are shown in parentheses.
(DOCX)

**S6 Table. Closest structural homologs of PacG-LBD as derived from a structural alignment using DALI (12) with all 3D structures deposited in the protein data bank.**
(DOCX)

**S7 Table. Strains and plasmids used in this study.**
(DOCX)

**S8 Table. Oligonucleotides used in this study.**
(DOCX)

**S9 Table. Sequences of proteins used in this study.** The sequence extension containing the hexa-histidine tag is shown in bold.
(DOCX)

## Acknowledgments

We are indebted to Dr. Mike Manson for constructive scientific comments and editing the manuscript. We are grateful to the Spanish Synchrotron Light Facility (ALBA) and the European Synchrotron Radiation Facility (ESRF) for providing diffraction time through proposals 2023087670 and MX2605, respectively.

## Author contributions

**Conceptualization:** Ashleigh Holmes, Jose A. Gavira, Miguel A. Matilla, Tino Krell.

**Data curation:** Ashleigh Holmes, Jose A. Gavira.

**Formal analysis:** Roberta Genova, Mario Cano-Muñoz.

**Funding acquisition:** Miguel A. Matilla, Tino Krell.

**Investigation:** Roberta Genova, Ashleigh Holmes, Mario Cano-Muñoz, Atsushi Ishihara, Naoki Ube, Taiji Nomura, Jose A. Gavira.

**Methodology:** Roberta Genova, Jose A. Gavira.

**Project administration:** Miguel A. Matilla, Tino Krell.

**Resources:** Atsushi Ishihara, Naoki Ube, Taiji Nomura.

**Supervision:** Miguel A. Matilla, Tino Krell.

**Validation:** Ashleigh Holmes, Jose A. Gavira, Miguel A. Matilla, Tino Krell.

**Visualization:** Mario Cano-Muñoz, Jose A. Gavira, Miguel A. Matilla, Tino Krell.

**Writing – original draft:** Roberta Genova, Ashleigh Holmes, Mario Cano-Muñoz, Atsushi Ishihara, Naoki Ube, Taiji Nomura, Jose A. Gavira, Miguel A. Matilla, Tino Krell.

**Writing – review & editing:** Miguel A. Matilla, Tino Krell.

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
