## [Decision Letter · Decision Letter 0]

24 Mar 2026

PPATHOGENS-D-26-00373

Chemotaxis to Plant Defense Compounds in Phytopathogens

PLOS Pathogens

Dear Dr. Krell,

Thank you for submitting your manuscript to PLOS Pathogens. After careful consideration, we feel that it has merit but does not fully meet PLOS Pathogens's publication criteria as it currently stands. Therefore, we invite you to submit a revised version of the manuscript that addresses the points raised during the review process.

We look forward to receiving your revised manuscript.

Kind regards,

Nian Wang

Academic Editor

PLOS Pathogens

Shou-Wei Ding

Section Editor

PLOS Pathogens

Sumita Bhaduri-McIntosh

Editor-in-Chief

PLOS Pathogens

orcid.org/0000-0003-2946-9497

Michael Malim

Editor-in-Chief

PLOS Pathogens

orcid.org/0000-0002-7699-2064

**Journal Requirements:**

1) Please provide an Author Summary. This should appear in your manuscript between the Abstract (if applicable) and the Introduction, and should be 150-200 words long. The aim should be to make your findings accessible to a wide audience that includes both scientists and non-scientists. Sample summaries can be found on our website under Submission Guidelines:

https://journals.plos.org/plospathogens/s/submission-guidelines#loc-parts-of-a-submission

2) We noticed that you used the phrase 'data not shown' in the manuscript. We do not allow these references, as the PLOS data access policy requires that all data be either published with the manuscript or made available in a publicly accessible database. Please amend the supplementary material to include the referenced data or remove the references.

- ® on page: 25

- TM on page: 23.

5) We have noticed that you have uploaded Supporting Information files, but you have not included a list of legends. Please add a full list of legends for your Supporting Information files after the references list.

7) Please ensure that the funders and grant numbers match between the Financial Disclosure field and the Funding Information tab in your submission form. Note that the funders must be provided in the same order in both places as well.                                                  ".

8) Please ensure to update your Data Availability Statement in the online submission form..

**Reviewers' Comments:**

Reviewer's Responses to Questions

**Part I - Summary**

Reviewer #1: The manuscript by Genova et al. investigates the chemotaxis response of the plant pathogen Pectobacterium atrosepticum. The authors identify important plant defense compounds as chemoreceptor ligands for three chemoreceptors and show a role of PacG in plant virulence. This is a well-written manuscript describing carefully composed and conducted experiments. The identification of specific chemoattractants and their corresponding chemoreceptors will contribute to a better understanding of plant pathogen/host interaction, which is an understudied area of research.

Reviewer #2: This is a revised mansucript for which I did not serve as an original reviewer. The authors seem to have addressed the reviewers comments. I have only one conceptual questions and additional editorial comments.

Reviewer #3: The paper by Genova et al. present the findings of studies on three plant pathogen chemoreceptors, terms PacG, PacH, PacI. The authors find the ligands of these chemoreceptors, and determine that they function in chemotaxis, and also are important for plant infection. The manuscript is well written and clear. Overall the work is compelling, impactful, and interesting. I have only a few comments. My only suggestions are slight to change some wording in the introduction about the idea that because chemoreceptors are found in plant pathogen, they must sense plant compounds. While a decent idea, it’s hypothetical so please tone down:

• Line 58 and 72 are too strong. E.g. at 58, change to “suggesting that many of these receptors might sense…”

Reviewer #4: In this manuscript, the authors investigated three chemoreceptors in Pectobacterium. They identified the ligands. They also conducted mutagenesis study. Overall, this study contains some interesting information. There are some major issues that must be addressed though.

**Part II – Major Issues: Key Experiments Required for Acceptance**

Reviewer #1: None.

Reviewer #2: Authors provide MIC for eahc of the predicted LBD ligands and also provide semi-quantitative capillary assays for chemotaxis responses usign arange of concentations. Despite the compounds being toxic at high concentrations, the positive chemotaxis responses suggest these are attractants - this makes sense sine they are chemotactically stimulating cells behavioural responses at concentrations far below the MIC for all compounds tested except for vanillin (Fig. S7). For vanillin, a positive chemotaxis response is detected for concentrations >MIC (6mM per Table 1 for PacH and Fig S7). The observation is not addressed despite its incongruence. While it makes senses that a phytopathogen could detect a potentially toxic compound as an attractant at concentrations far below the MIC, for this attractant, it is not clear what would be the advantage to sense a chemattractant at a concentration that can kill the cells. Shouldn't the results suggest that chemotaxis attraction to vanillin/LBD binding to vanillin may not be physiologically relevant? Other issues?

Reviewer #3: None, it's good

Reviewer #4: 1. Statistical analysis was missing for data related to Fig. 4 and Fig. 5B

2. Line 226, the authors stated: To monitor chemotaxis of SCRI1043 to the ligands identified above, we conducted quantitative capillary chemotaxis assays. Initial experiments conducted in standard conditions, involving bacterial cell culture in minimal medium supplemented with glucose as sole carbon source, did not show any taxis to vanillin, salicylate or benzoate (data not shown). Chemotaxis can sometimes be induced by the presence of the cognate ligands in the culture medium [57], so we next monitored chemotaxis to vanillin, salicylate and benzoate in cultures grown in minimal medium supplemented with vanillin or benzoate.

Based on the aforementioned writing, the authors need to clearly state what was included in Fig. 4 figure legends for each experiment.

3. For the virulence assays, please include pictures of infected roots and leaves in supplementary figures.

4. It is required to conduct complementation assay for pacG mutant and pacHIG mutant.

**Part III – Minor Issues: Editorial and Data Presentation Modifications**

Reviewer #1: None.

Reviewer #2: 1. Fig. 1 legend has not been revised from the previous version and include innformation that have been moved to the SI (e.g. reference to red brackets and a panel B)

2. Fig S7- please add labels A, B and C to the individal histogram graphs since the figure caption refers to them.

Reviewer #3: See above

Reviewer #4: 1.Fig. 1. It listed A), but there is no A) in the figure. This figure quality needs to improve.

PLOS authors have the option to publish the peer review history of their article (what does this mean?). If published, this will include your full peer review and any attached files.

Reviewer #1: No

Reviewer #2: No

Reviewer #3: No

Reviewer #4: No

**Figure resubmission:**
---

## [Editor Report · Decision Letter 1]

4 May 2026

Dear Dr. Krell,

We are pleased to inform you that your manuscript 'Chemotaxis to Plant Defense Compounds in Phytopathogens' has been provisionally accepted for publication in PLOS Pathogens.

Please correct the following in your proof stage:

Please provide the statistical method for Fig. 5B legend.

Best regards,

Nian Wang

Academic Editor

PLOS Pathogens

Shou-Wei Ding

Section Editor

PLOS Pathogens

Sumita Bhaduri-McIntosh

Editor-in-Chief

PLOS Pathogens

orcid.org/0000-0003-2946-9497

Michael Malim

Editor-in-Chief

PLOS Pathogens

orcid.org/0000-0002-7699-2064
---

## [Editor Report · Acceptance letter]

Dear Dr. Krell,

We are delighted to inform you that your manuscript, "Chemotaxis to Plant Defense Compounds in Phytopathogens," has been formally accepted for publication in PLOS Pathogens.

Best regards,

Sumita Bhaduri-McIntosh

Editor-in-Chief

PLOS Pathogens

orcid.org/0000-0003-2946-9497

Michael Malim

Editor-in-Chief

PLOS Pathogens

orcid.org/0000-0002-7699-2064